# The Effects of a Physical Activity, Nutrition, and Body Image Intervention on Girls in Intermediate Schools in Saudi Arabia

**DOI:** 10.3390/ijerph191811314

**Published:** 2022-09-08

**Authors:** Abeer Ahmad Bahathig, Hazizi Abu Saad

**Affiliations:** 1Department of Nutrition, Faculty of Medicine and Health Sciences, Universiti Putra Malaysia, Serdang 43400, Malaysia; 2Department of Nutrition and Food Science, College of Home Economics, Northern Border University, Arar 91431, Saudi Arabia

**Keywords:** physical activity, nutrition, body image, intervention, school students

## Abstract

Background: This cluster-randomized study aimed to evaluate the effects of an intervention focused on physical activity, nutrition, and body image perception among girls in intermediate schools in Saudi Arabia. A seminar was delivered to the mothers of the girls in the experimental group. The experimental group then attended six interactive sessions within 3 months featuring physical activity, nutrition, and body image perception, followed by a 3-month follow-up period. A total of 138 respondents (68 in the experimental group and 70 in the control group) completed the intervention. Each participant’s body mass index-for-age z-score and waist circumference were measured, and they completed a physical activity, sedentary behavior, and body image perception questionnaires before and after the intervention and at the follow-up. The intervention was evaluated using within- and between-groups generalized estimating equations. There were no significant changes in the respondents’ body mass index-for-age z-score or waist circumference (*p* > 0.05). However, immediate significant improvements were seen in physical activity, sedentary behavior, and body image satisfaction among the experimental group, and these improvements remained at follow-up (*p* < 0.001). These differences were more significant among the experimental group than among the control group. We found this intervention effective. Future studies can adapt this intervention for adolescent boys and extend its duration to improve the body mass index outcome.

## 1. Introduction

Physical inactivity can increase the risk of diabetes, hypertension, heart diseases, stroke, muscular and mental problems, and several types of cancer [1]. Nevertheless, 73.7% of Saudi females do not achieve adequate daily levels of physical activity (PA) [2]. This is important, as a lack of activity is linked to increases in body mass index (BMI), waist circumference (WC), and obesity-related diseases [3]. Furthermore, 23.3% of normal-weight respondents in Saudi Arabia, especially girls, spend more than three hours on screens on weekdays [4]. Moreover, body image (BI) dissatisfaction (BID) may be a risk factor for high-level alcohol and drug consumption and self-harm [5]. According to Albawardi et al. [6], 87% of Saudi adolescent girls surveyed had poor BI. BID can be caused by a lack of PA [7]. Adolescents who have healthy BIs at young ages are at decreased risk of unhealthy behaviors in the future [5]. Therefore, interventions promoting positive BI can reduce obesity.

Darabi et al. [8] confirmed that providing adolescents with information regarding PA during educational interventions can help provide them with the necessary skills to practice PA. Moreover, interventions including nutritional knowledge can improve PA habits [9]. One study provided adolescents with an educational intervention that was based on the social cognitive theory (SCT) and focused on PA information and effects, the drawbacks of high levels of sedentary behaviors (SBs; e.g., screen time), and the benefits of having a normal body weight. After the intervention, adolescents demonstrated improved PA levels and decreased SB and BMIs [10]. Another study based on SCT also showed that adolescents’ BMIs and WCs decreased after three months of educational sessions focused on PA and nutrition [11].

Furthermore, school is the optimal place to educate adolescents [12] because the setting provides facilities, curricula, and classrooms [11]. The most successful educational methods have been used in school settings to develop PA behavior and reduce BMIs in adolescents [13]. Students develop their behaviors during the school year, and ages 12 to 17 are best suited for interventions at school [14]. This means that behavior is more likely to change in adolescence than in adulthood. Therefore, the purpose of this study was to evaluate the effects of a PA, nutrition, and BI intervention on girls in intermediate schools in Saudi Arabia. Here, we report the results of our 12-week intervention study on the PA, nutrition, and BID of adolescent girls in Arar, Saudi Arabia. We also assessed the follow-up effects of the intervention 3 months after the intervention ended.

## 2. Materials and Methods

### 2.1. Study Design and Sample

A cluster-randomized controlled trial was conducted from January to June 2020 in Arar, Saudi Arabia. There are 22 government intermediate schools for girls in Saudi Arabia, and 2 of these schools participated in the present study. Students from one school received the intervention and comprised the experimental group (EG), and students from the second school served as the control group (CG). To avoid contamination, two different schools were selected, and these two schools were separated by a reasonable geographical distance. The selection and allocation of both the schools and the respondents were randomly conducted by the first author using Excel software. This study was open to Saudi girls that could practice PA between 13 and 14 years old; each participant provided a consent form and their parents’ approval to participate, and the mothers of the girls in the EG agreed to participate as well.

Rosner’s [15] formula, based on the mean change in knowledge for the EG and the CG [11], was used to calculate the minimum sample size for each group (68 respondents/group). For withdrawals, 15% was added [16]. The final sample size needed for the two groups was 160 respondents:(1)$$n\frac{\left(\sigma_{1}^{2}+\sigma_{2}^{2}/k\right){\left(z_{1-\alpha /2}+Z_{1-\beta }\right)}^{2}}{\Delta^{2}}$$
where:

*n* = sample size per group; *σ*_1_ = standard deviation (EG); *σ*_2_ = standard deviation (CG); *Δ* = EG mean change − CG mean change; κ ratio = *n*2/*n*1 = 1; *Z*_1−*α*/2_ = 1.96 (confidence interval 95%) (where *α* = 0.05); and *Z*_1−*β*_ = 0.842 (standard value of a normal distribution).

### 2.2. Ethical Approval

This study received ethical approval from the following institutions: (1) Universiti Putra Malaysia, Ethics Committee for Research Involving Human Subjects, JKEUPM (UPM/TNCPI/RMC/JKEUPM/1.4.18.2 (JKEUPM)), Selangor, Malaysia; (2) Northern Border University, The Local Committee of Bio-Ethics HAP-09-A-043 (13/40/H); and (3) the Ministry of Education in Arar, Saudi Arabia. All consent forms from the respondents and their parents were collected before data collection began. The respondents understood the aim and procedures of this intervention. All respondents’ information was kept confidential.

### 2.3. Educational Intervention

The focus of the current intervention was to improve the respondents’ PA, SB, and BI behaviors, BMI-for-age z-score (BAZ), and WC to enhance the health of girls aged 13 to 14 years in Arar. The intervention was designed based on a previous needs assessment and prior studies [11,17,18,19,20].

This educational intervention was implemented among the EG after being reviewed by six specialized experts from Arar and was modified based on their feedback. Some important points regarding the intervention are as follows:The mothers of the girls in the EG received information through a 60 min seminar before the education of the EG. The seminar focused on the importance of PA, the risks of physical inactivity, the risks of SB, the health effects of prolonged SB, how to replace SB with PA, healthy eating, and positive BI.The EG received six educational sessions on PA, nutrition, and BI during one academic semester. The sessions explained the definition, levels, advantages, and daily recommendations for PA and the importance of walking and practicing PA anytime and anywhere. The sessions also covered the disadvantages of SB and extended screen time and how to change SB to activity. The intervention addressed the benefits of BI satisfaction (BIS) and the drawbacks of BID. The intervention also covered obesity, type 2 diabetes, dietary guidelines for Saudis, food groups, serving sizes, food habits, portion size, food labels, dietary recommendations, and body weight status.Each 90 min session was conducted once every two weeks (for a total of three months). Each session consisted of a knowledge discussion and an activities section. Thus, the respondents acquired skills to help them practice and maintain new behaviors. The EG group was divided into two cohorts because of the capacity of the classroom; half of the EG received the session on one day, and the other half received the session the next day.Teaching aids, such as PowerPoint presentations, booklets, games, papers and cards, school boards, group discussions, and stickers, were used in the intervention. The CG received their regular education during the study intervention.From a total of 540 min of intervention time, 270 min were spent for educational lessons: nutrition (160 min), physical activity (95 min), and body image perception (15 min).Another 270 min were spent in practical activities: nutrition (100 min), physical activity (155 min), and body image perception (15 min).Examples of practical activities included respondents matching food groups with The Healthy Food Palm [21], comparing between two food labels, warm-up exercises, skipping, walking, dancing using the hoop, jogging, etc.

### 2.4. Data Collection and Measures

The self-reported data collection took place at the three time points: pre-intervention (baseline), post-intervention, and follow-up (3 months after the post-intervention).

### 2.5. Research Instruments

#### 2.5.1. Respondents’ Anthropometries

Double measurements were conducted to ensure accuracy. The respondents wore light clothes with empty pockets without shoes with heads upright and stood on a straight stand to measure their body weight (to the nearest 0.1 kg) and height (to the nearest 0.1 cm) using a Detecto solo digital scale, Digital Clinical Scale, Webb City, MO (USA). The BMI-for-age z-score (BAZ) was measured according to the growth reference for (girls) aged 5 to 19 years old, as follows: <–2 standard deviations (SD) = thin; ≥–2 SD and ≤1 SD = normal weight; >+1 SD = overweight; and >+2 SD = obese [22]. Each participant’s WC was measured between her pelvis and rib cage, using a measuring tape, to the nearest 0.1 cm, using the following cut-off points: 72.3 cm = overweight and 77 cm = obese [23].

#### 2.5.2. Physical Activity

The Physical Activity Questionnaire for Older Children (PAQ-C) was used to assess the respondents’ PA levels during the previous seven days [24]. The Cronbach’s alpha for the PAQ-C was previously reported to be 0.777 [25]. The PAQ-C contains nine items, and the mean score of all items represents the respondent’s PA score. The PAQ-C uses a five-point Likert scale, with 1 indicating low PA and 5 indicating high PA. Item 1 addresses PA in the respondent’s spare time, with several activities scored from 1 to 5 (low to high) and the mean score representing the individual’s overall score for this item. The activities in this item were modified to fit Saudi society based on expert guidance from three specialists from Arar. For example, listed activities such as ice skating, cross-country skiing, and ice hockey/ringette were removed and replaced with other relevant and popular activities for Saudi girls, such as game chairs, dancing with the ring, pull the cord, etc. Item 9 contains activities scored from 1 (none) to 5 (very often). Items 2 to 8 include activities scored from 1 to 5 [26]. The total mean PA scores are categorized as follows: ≤2.3 = low, 2.4 to 3.7 = moderate, and ≥3.8 = high [27].

#### 2.5.3. Sedentary Behavior

The Adolescent Sedentary Activity Questionnaire (ASAQ) was utilized to analyze the respondents’ sedentary behavior habits. The ASAQ contains 11 items to evaluate the time spent on SB and has a test–retest reliability of ≥0.70 [27]. The respondents were asked about the time they spent performing the following five activities: (1) screen time (watching TV, watching videos and DVDs, using a computer for fun, using an iPad or smartphone for fun, and playing video games and computer games); (2) education (studying without a computer, studying with a computer, and being tutored); (3) travel (plane, car, and train); (4) cultural activities (reading for fun, doing crafts or hobbies, and practicing a musical instrument); and (5) activities (using the phone, chilling, chatting with friends, and going to the mosque). The total mean of SB time on weekdays or weekends is the average time spent on all categories [28]. The respondents’ ASAQ levels were classified based on Asare et al.’s [29] definitions (≥4 h/day = high level; <4 h/day = low level).

#### 2.5.4. Body Image Satisfaction

The Stunkard Figure Rating Scale was used to assess the respondents’ BIS. With a test–retest reliability of 0.8, the scale includes nine images, classified from thinnest (1) to heaviest (9) [30,31]. Each respondent selected one figure that she thought represented her current BI, one that represented her desired BI, and one that represented her ideal BI. The difference between the number of each participant’s desired BI figure and the number of her current BI figure indicated her BIS. If the difference was zero, she was satisfied with her BI; if the difference was not zero, she was dissatisfied with her BI [32].

### 2.6. Statistical Analysis

SPSS software (version 25; SPSS, IBM, Chicago, IL, USA) was used to analyze the data. Means and standard deviations (SDs) were used for continuous variables, and percentages and frequencies were used for categorical variables. To determine the effects of the intervention, the results were analyzed using per protocol analysis. A generalized estimating equation (GEE) was used to evaluate the effectiveness of the present intervention on anthropometric measurements (BAZ and WC), PA, SB, and BIS over the study period within and between groups. A chi-square test was used to determine the associations between categorical variables, and the post hoc Bonferroni test was used to determine the differences in each group. The related-samples Cochran’s Q test was used to evaluate the differences in BIS for both groups over time. The significance value was set at *p* < 0.05.

## 3. Results

From a total of 160 respondents, 68 (89.4%) from the EG and 70 (89.7%) from the CG completed the intervention study. Table 1 presents the respondents’ descriptive statistics (BAZ, WC, PA and screen time and total SB on weekdays and weekends) over the study period. The percentage of overweight and obese respondents in the EG (based on BAZ) decreased slightly, from 22.1% to 20.6%. The percentage of normal-weight respondents increased, from 63.2% to 64.7%. The proportions for each category were similar among the CG. Neither group demonstrated a change in WC throughout the study period. The percentage of respondents with low PA levels decreased from 100% (pre-intervention) to 19.4% (follow-up), and the percentage of respondents with moderate PA levels increased from 80.9% (pre-intervention) to 82.4% (follow-up). In the CG, the percentages of respondents with low and moderate PA levels did not change throughout the study period.

In the EG, the proportions of respondents with low levels of screen time during the weekdays and the weekends both increased (from 11.8% to 58.8% on weekdays and from 7.4% to 22.1% on weekends). In contrast, the proportions of respondents with high levels of screen time during the weekdays and weekends both decreased (from 88.2% to 41.2% on weekdays and from 92.6% to 77.9% on weekends). In the CG, the percentage of respondents with low levels of screen time during the weekdays decreased slightly, from 7.1% to 2.9%; the percentage of those with high levels of screen time during the weekdays increased, from 92.9% to 97.1%. The percentage of those with high levels of screen time during the weekends remained stable at 100%. In the EG, the percentages of respondents who reported low levels of total SB during the weekdays increased throughout the study period, from 2.9% to 16.2% (weekdays) and from 1.5% to 13.2% (weekends). The percentages of those who reported high SB levels decreased throughout the study period, from 97.1% to 83.8% (weekdays) and from 98.5% to 86.8% (weekends). In the CG, the percentages of respondents who reported high levels of SB increased, from 98.6% to 100% (weekdays and weekends).

The GEE test was used to measure the effects of this intervention on the respondents’ BAZ, WC, PA, screen time (weekdays and weekends), and total SB (weekdays and weekends). The scores for BAZ, PA, total SB on weekdays, and screen time on weekends were included as covariates in the analysis because of the significant pre-intervention between-group differences; thus, the post-test and follow-up scores were indicative of the changes within and between groups. The descriptions of these variables over time for both groups are shown in Figure 1.

The results of the GEE test are presented in Table 2. The time effect significantly differed among the respondents: BAZ (χ^2^ = 38.15, *p* = 0.001); WC (χ^2^ = 6.504, *p* = 0.039); PA (χ^2^ = 1114.041, *p* < 0.001); screen time during weekdays (χ^2^ = 48.67, *p* < 0.001); total SB during weekdays (χ^2^ = 52.11; *p* < 0.001); screen time during weekends (χ^2^ = 91.814, *p* < 0.001); total SB during weekends (χ^2^ = 23.05, *p* < 0.001). Differences were not significant based on the effect of the group: BAZ (χ^2^ = 0.164, *p* = 0.686) or WC (χ^2^ = 2.23, *p* = 0.135). They were significantly different for the following parameters: PA (χ^2^ = 588.588, *p* < 0.001); screen time during weekdays (χ^2^ = 3880.014, *p* < 0.001); total SB during weekdays (χ^2^ = 39.028, *p* < 0.001); screen time during weekends (χ^2^ = 116.762; *p* < 0.001); and total SB during weekends (χ^2^ = 70.557, *p* < 0.001). The interaction (time x group) effect was not significantly different for BAZ (χ^2^ = 1.353; *p* = 0.508) or WC (χ^2^ = 0.774, *p* = 0.679), indicating that the two groups had the same BAZ and WC patterns. Furthermore, there were significant changes for PA (χ^2^ = 735.809, *p* < 0.001); screen time during weekdays (χ^2^ = 95.93, *p* < 0.001); total SB during weekdays (χ^2^ = 32.76, *p* < 0.001); screen time during weekends (χ^2^ = 30.688, *p* < 0.001); and total SB during weekends (χ^2^ = 35.923, *p* < 0.001). Thus, the EG and CG had different patterns for PA, screen time, and total SB.

The post hoc Bonferroni test was used to determine the differences in each group (Table 3). For BAZ, there were significant changes between the pre- and post-intervention values and between the post-intervention and follow-up values in the EG (*p* < 0.001), while there was no significant change among the CG from pre-intervention to follow-up (*p* > 0.05). For WC, neither the EG nor the CG significantly differed from pre-intervention to follow-up (*p* = 1.0 for both). In the EG, for PA, there was a significant difference between the mean pre-intervention and post-intervention values (*p* < 0.001), between the mean pre-intervention and follow-up values (*p* < 0.001), and between the mean post-intervention and follow-up values (*p* = 0.05). In the CG, there were also significant differences between all three measurement points (*p* < 0.05 for all). Regarding the change in screen time on weekdays, there were significant differences in the EG between the pre-intervention and post-intervention values (*p* < 0.001) and between the pre-intervention and follow-up values (*p* < 0.001) but not between the post-intervention and follow-up values (*p* = 0.22). In the CG, there was a significant difference between the pre-intervention and post-intervention values (*p* = 0.01) but not between the pre-intervention and follow-up values (*p* = 0.68) or between the post-intervention and follow-up values (*p* = 0.36). In the EG, for the total SB time on weekdays, significant differences were reported between the pre-intervention and post-intervention values and between the pre-intervention and follow-up values (*p* < 0.001 for both) but not between the post-intervention and follow-up values (*p* = 0.31). In the CG, there was a significant change between the pre-intervention and follow-up values (*p* = 0.03) but not between the post-intervention and follow-up values (*p* = 0.22). In the EG, there were significant changes in weekend screen time between pre-intervention and post-intervention and between pre-intervention and follow-up (*p* < 0.001 for both) but not between post-intervention and follow-up (*p* = 0.61). In the CG, a significant change was found between baseline and post-intervention (*p* = 0.01) but not between baseline and follow-up (*p* = 0.66) or between post-intervention and follow-up (*p* = 0.27). For the total SB time on weekends, in the EG, the differences between pre-intervention and post-intervention and between pre-intervention and follow-up were significantly different (*p* < 0.001 for both), while that between post-intervention and follow-up was not significant (*p* = 0.32). In the CG, the changes between baseline and post-intervention and between post-intervention and follow-up were not significant (*p* > 0.05 for both), while that between baseline and follow-up was significant (*p* = 0.01).

The Cohen [33] effect size of the time for both groups was small for BAZ (EG, *d* = 0.02; CG, *d* = 0.01) and WC (EG, *d* = 0.05; CG, *d* = 0.01). A large effect size of time was found in the EG (*d* = 4.46); it was four times higher than that in the CG (*d* = 0.92). A large effect size of time was found on weekday screen time in the EG (*d* = 1.38), but this effect size was small in the CG (*d* = 0.20). For the total SB, the effect size of time in the EG was large (*d* = 0.91); it was medium in the CG (*d* = 0.45). The effect size of time on screen time during weekends was large in the EG (*d* = 0.98) and small in the CG (*d* = 0.16). For the total SB, the effect size of time was large (*d* = 0.75) in the EG and medium (*d* = 0.52) in the CG.

The Bonferroni test was used to compare the respondents’ variables between groups at different time points (Table 4). There was no significant difference between the two groups for BAZ at pre-intervention (*p* = 1.0), at follow-up (*p* = 1.0), or at post-intervention (*p* = 0.28). With respect to WC, the results revealed no significant differences between the two groups at any of the three time points (*p* = 1.0 for all). The adjusted PA was significantly different between the groups at pre-intervention (*p* = 0.01). The differences between the groups were significant at post-intervention and follow-up (*p* < 0.001). With respect to weekday screen time, there was no significant difference between the groups at pre-intervention (*p* = 0.90), but at post-intervention and follow-up the differences were significant (*p* < 0.001 at both time points).

For total SB, no significant change was found at baseline (*p* = 0.83); at post-intervention and follow-up, the differences were significant (*p* < 0.001 for both). With respect to weekend screen time, there were significant differences between the groups at baseline, post-intervention, and follow-up (*p* < 0.001 for all). For total SB, the difference was insignificant at baseline (*p* = 0.08) but significant at post-intervention and follow-up (*p* < 0.001).

The Cohen [32] effect sizes over the study times between the groups are presented in Table 4. With regard to BAZ, the effect size was small at all three time points (pre-intervention, *d* = 0.35; post-intervention, *d* = 0.34; follow-up, *d* = 0.36). It was also small at all three time points for WC (pre-intervention, *d* = 0.22; post-intervention, *d* = 0.25; follow-up, *d* = 0.26). With respect to PA, the effect size was large at all three time points (pre-intervention, *d* = 0.87; post-intervention, *d* = 4.31; follow-up, *d* = 4.97). With respect to weekday screen time, the effect size was medium at post-intervention (*d* = 0.68), large at follow-up (*d* =1.37), and small at baseline (*d* = 0.02). For the total SB, the effect size was small at baseline (*d* = 0.18), medium at post-intervention (*d* = 0.45), and large at follow-up (*d* = 1.09). The effect size of weekend screen time was large at post-intervention (*d* = 1.15) and follow-up (*d* = 1.69) and medium at pre-intervention (*d* = 0.61). For the total SB, the effect size was small at baseline (*d* = 0.39) and large at post-intervention (*d* = 0.80) and follow-up (*d* = 1.38).

The related-samples Cochran’s Q test was used to evaluate the differences in BIS for both groups over time (Table 5). There was a significant difference among the EG respondents in BIS over time (*p* < 0.001), but the same was not true among the CG respondents (*p* = 0.247). Moreover, significant differences were reported for BIS at post-intervention and follow-up between EG and CG (*p* = 0.002 and *p* = 0.006, respectively). The proportions of respondents in the EG with BIS increased (from 14.5% to 51.3%). In contrast, the proportions of respondents in the CG decreased (from 30.8% to 29.5%) over time.

## 4. Discussion

Practicing PA and reducing SB can prevent obesity and non-communicable diseases [1]. This educational intervention focused on PA, nutrition, and BI perception; it was associated with significant improvements in PA, SB, and BIS among EG girls aged 13 to 14 years in Arar, Saudi Arabia, after they completed the six interactive sessions (*p* < 0.001), and the respondents maintained these improvements at the three-month follow-up (*p* = 0.001). The improvement among the EG respondents was greater than that among the CG respondents (*p* = 0.001).

These results are in line with previous findings [10,12,19]. Bagherniya et al. [10] conducted a randomized study among adolescents from Iran. After the educational program on PA and nutrition, the PA duration increased significantly among respondents in the intervention group (*p* < 0.001). These findings were consistent with the results of Hefni’s [19] study, which evaluated a nutritional intervention among Saudi adolescent girls and reported that, compared to baseline, the number of girls participating in PA significantly increased after 3 months of intervention (*p* = 0.001). Numerous studies have demonstrated the effectiveness of interventions based on SCT to improve PA at post-test and follow-up time points among adolescents (*p* < 0.05) [10,12,19]. Therefore, PA interventions effectively contribute to the improvement in PA among adolescents.

In this study, the EG respondents’ weekend and weekday screen time and SB were lower after the intervention; these differences were bigger in the EG than in the CG (*p* < 0.001). The reduced screen time and SB may have been results of the respondents’ increased levels of PA. In a study by Salem and Said [9], adolescents spent significantly less time watching television and using the internet immediately after an educational intervention, and this difference remained at the 12-week follow-up (*p* < 0.001). Furthermore, Bagherniya et al. [10] and Hefni [19] evaluated face-to-face interventions, which were implemented within short- and long-term interventions among adolescents at their schools, and found significant decreases in SB and screen time after the intervention. Therefore, sessions that aim to reduce prolonged time spent on SB and increase active time among school students are effective.

In this study, the EG respondents’ BIS increased significantly after six 90 min sessions; this increase among the EG respondents was larger than that among the CG respondents. Unhappiness with BI can harm girls’ psychological and physical health; in Dunstan et al.’s [34] study, the respondents who attended the intervention demonstrated reduced BID—significantly more than those who did not complete the intervention. The present study revealed similar results. Dunstan et al. [34] reported that BID was significantly lower after six weeks of intervention compared to baseline (*p* < 0.01). These results, along with the findings of Annesi et al. [35] and Richardson and Paxton [36], indicate that providing educational interventions for school students can enhance BIS. The present study intervention had a combined focus on the importance of PA, BI acceptance, and nutrition, and it yielded positive results.

This study found that the BAZ and WC changed significantly based on effect time (*p* = 0.001 and *p* = 0.039, respectively). However, the BAZ and WC patterns over the study period were similar for both groups. Therefore, the interaction of (time × group) was not significant for BAZ (*p* = 0.508) or WC (*p* = 0.679); this nutrition intervention did not affect BAZ and WC significantly. The present findings are in line with those from Pbert et al. [37] and Sharif Ishak et al. [38]; their interventions among adolescents who attended educational sessions did not result in any significant changes. Other studies showed positive differences in reducing or maintaining BMI or WC after their interventions [11,19,39]. Such differences in results may be attributed to the duration or content of the intervention or the age of the respondents. Older children have a greater interest in their health than younger children [39]. Including mixed-weight respondents in research can slightly change the BMI [40]; however, this study aimed to develop PA, SB, and positive BI. The intervention period lasted 3 months, followed by a 3-month follow-up, which may be considered a short length of time to bring about changes in BMI. Whitlock et al. [41] stated that a period of 6 months or more is required to analyze changes in body weight. The current intervention was implemented in a shorter time, which may explain why the intervention did not affect BAZ or WC. A longer intervention period may yield significant changes in BAZ and WC.

Some limitations were identified in this study. First, the study respondents were from two schools; the experimental group came from one school, and the control group came from the other. Therefore, limited generalizations can be made from the findings. The researchers could not implement the intervention in additional schools due to the lack of time and possibilities. Second, this intervention did not apply to boys; schools in Saudi Arabia do not include both sexes together. Third, a self-reported questionnaire was used for data collection, which may have been affected by response bias. However, this is common in educational interventions. One strength of the present study was its methodology, which was developed in accordance with the needs of the study population; furthermore, the respondents and schools were randomly selected. A suitable sample size was included for the study design. At the pre-intervention time point, all significant differences were adjusted for multiple comparisons. To assess BI perception, figures were used rather than questions to improve ease and minimize bias in the data collection.

## 5. Conclusions

The present educational intervention demonstrated significant positive impacts on PA, SB, and BIS in girls aged 13 and 14 years at the post-intervention and follow-up time points (*p* < 0.05). To the best of our knowledge, this is the first study to provide data using the specific variables of BAZ, WC, PA, SB, and BIS and an educational program aimed at improving PA, SB, and BIS. This intervention design can be used by The Ministry of Education in Arar to provide sessions and workshops aimed at improving adolescents’ lifestyles. The Ministry of Health can also prepare outdoor activities and organize awareness seminars regarding PA and BI to improve the health of adolescents. Although this intervention was not associated with significant decreases in BAZ and WC, it was also not associated with increases in these parameters. This indicates the efficacy of the current intervention. However, a prolonged intervention may yield better results on BMI and WC; future studies can take this into account. Interventions for PA, SB, BIS, and obesity prevention for adolescent girls can be carried out with the help of trained influential students who support their peers in their year group. Future studies can modify this intervention for use among boys.

## Figures and Tables

**Figure 1 ijerph-19-11314-f001:**
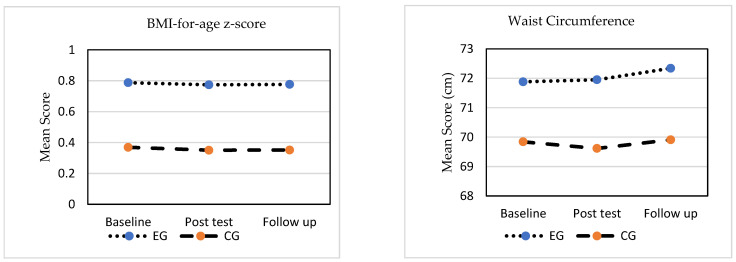
BMI-for-age z-score, waist circumference, physical activity, and screen time and total sedentary behavior on weekdays and weekends for both groups over time.

**Table 1 ijerph-19-11314-t001:** Descriptive Statistics of BMI-for-age z-score, waist circumference, physical activity and screen time and the total sedentary behavior on weekdays and weekends for both groups over time.

**Groups**	**Variables**		**Pre-Intervention** ***n* (%)**			**Post-Intervention** ***n* (%)**			**Follow-Up** ***n* (%)**	
	**Level of BMI-for-age z-score (BAZ)**									
**EG**	Thinness (<−2) SD		0 (0.0)			0 (0.0)			0 (0.0)	
	Normal (≥−2 to ≤+1) SD		43 (63.2)			43 (63.2)			44 (64.7)	
	Overweight (>+1) SD		15 (22.1)			14 (20.6)			14 (20.6)	
	Obese (>+2) SD		10 (14.7)			11 (16.2)			10 (14.7)	
**CG**	Thinness (<−2) SD		0 (0.0)			0 (0.0)			0 (0.0)	
	Normal (≥−2 to ≤+1) SD		53 (75.7)			53 (75.7)			53 (75.7)	
	Overweight (>+1) SD		11 (15.7)			11 (15.7)			11 (15.7)	
	Obese (>+2) SD		6 (8.6)			6 (8.6)			6 (8.6)	
	**Level of waist circumference**									
**EG**	Normal		37 (54.4)			37 (54.4)			37 (54.4)	
	Overweight (≥72.3) cm		16 (23.5)			16 (23.5)			16 (23.5)	
	Abdominal obesity (≥77) cm		15 (22.1)			15 (22.1)			15 (22.1)	
**CG**	Normal		47 (67.1)			47 (67.1)			47 (67.1)	
	Overweight (≥72.3) cm		14 (20)			14 (20)			14 (20)	
	Abdominal obesity (≥77) cm		9 (12.9)			9 (12.9)			9 (12.9)	
	**Physical Activity Levels**									
**EG**	Low ≤2.3		68 (100.0)			13 (19.7)			12 (19.4)	
	Moderate 2.4–3.7		0 (0.0)			55 (80.9)			56 (82.4)	
	High ≥3.8		0 (0.0)			0 (0.0)			0 (0.0)	
**CG**	Low ≤2.3		70 (100.0)			70 (100.0)			70 (100.0)	
	Moderate 2.4–3.7		0 (0.0)			0 (0.0)			0 (0.0)	
	High ≥3.8		0 (0.0)			0 (0.0)			0 (0.0)	
	**Sedentary Behavior**	**Weekdays**	**Weekends**		**Weekdays**		**Weekends**		**Weekdays**	**Weekends**
	**Level of Screen Time**									
**EG**	Low (≤2 h/day)	8 (11.8)	5 (7.4)		23 (33.8)		10 (14.7)		40 (58.8)	15 (22.1)
	High (>2 h/day)	60 (88.2)	63 (92.6)		45 (66.2)		58 (85.3)		28 (41.2)	53 (77.9)
**CG**	Low (≤2 h/day)	5 (7.1)	0 (0.0)		4 (5.7)		0 (0.0)		2 (2.9)	0 (0.0)
	High (>2 h/day)	65 (92.9)	70 (100.0)		66 (94.3)		70 (100.0)		68 (97.1)	70 (100.0)
	**Level of Total SB Time**									
**EG**	Low (<4 h/day)	2 (2.9)	1 (1.5)		4 (5.9)		1 (1.5)		11 (16.2)	9 (13.2)
	High (≥4 h/day)	66 (97.1)	67 (98.5)		64 (94.1)		67 (98.5)		57 (83.8)	59 (86.8)
**CG**	Low (<4 h/day)	1 (1.4)	1 (1.4)		0 (0.0)		0 (0.0)		0 (0.0)	0 (0.0)
	High (≥4 h/day)	69 (98.6)	69 (98.6)		70 (100.0)		70 (100.0)		70 (100.0)	70 (100.0)

*n* = Number of respondents, BMI = Body mass index, EG = The experimental group (68), CG = The control group (70).

**Table 2 ijerph-19-11314-t002:** Results of BMI-for-age z-score, waist circumference, physical activity, and screen time and the total sedentary behavior on weekdays and weekends based on generalized estimating equations.

Variables	Source	Wald Chi-Square	df	*p*-Value
**BAZ ^a^**	Time	38.15 *	2	<0.001
	Group	0.164	1	0.686
	Time x Group	1.353	2	0.508
	BAZ_ pre-intervention	12,675.6 *	1	<0.001
**WC**	Time	6.504 *	2	0.039
	Group	2.23	1	0.135
	Time x Group	0.774	2	0.679
**PAQ ^a^**	Time	1114.041 *	2	<0.001
	Group	588.588 *	1	<0.001
	Time x Group	735.809 *	2	<0.001
	PAQ_pre-intervention	136.958 *	1	<0.001
**Screen time (weekdays)**	Time	48.67 *	1	<0.001
	Group	3880.01 *	1	<0.001
	Time * Group	95.93 *	2	<0.001
**Total SB time ^a^ (weekdays)**	Time	52.11 *	1	<0.001
	Group	39.028 *	1	<0.001
	Time x Group	32.76 *	2	<0.001
	Total SB time_ pre-intervention	45.37 *	2	<0.001
**Screen time ^a^ (weekends)**	Time	91.814 *	2	<0.001
	Group	116.762 *	1	<0.001
	Time x Group	30.688 *	2	<0.001
	Screen time_pre-intervention	276.911 *	1	<0.001
**Total SB time**	Time	23.05 *	2	<0.001
	Group	70.557 *	1	<0.001
	Time x Group	35.923 *	2	<0.001

df = Degrees of freedom, * significant score at (*p* < 0.05), ^a^ = adjusted mean difference: pre-intervention score considered as a covariate.

**Table 3 ijerph-19-11314-t003:** Pairwise Comparison of BMI-for-age z-score, waist circumference, physical activity, and screen time and total sedentary behavior on weekdays and weekends, mean score over time for both groups.

Variables	Group	Test	Test	Mean Difference (I–J)	SE	*p*-Value	95% CI for Difference		*d*
							LB	UB	
**BAZ ^a^**	**EG**	Pre-intervention	Post-intervention	0.0770 *	0.01	<0.001	0.04	0.12	0.02
		Pre-intervention	Follow-up	−0.01	0.02	1.00	−0.06	0.04	
		Post-intervention	Follow-up	−0.0732	0.01	<0.001	−0.11	−0.03	
	**CG**	Pre-intervention	Post-intervention	0.05	0.03	0.87	−0.03	0.13	0.01
		Pre-intervention	Follow-up	0.00	0.00	1.00	−0.01	0.01	
		Post-intervention	Follow-up	−0.01	0.03	1.00	−0.06	0.05	
**WC**	**EG**	Pre-intervention	Post-intervention	−0.45	0.31	1.00	−1.28	0.39	0.05
		Pre-intervention	Follow-up	2.11	1.51	1.00	−1.88	6.10	
		Post-intervention	Follow-up	2.50	1.54	1.00	−1.80	6.80	
	**CG**	Pre-intervention	Post-intervention	−0.07	0.32	1.00	−0.73	0.58	0.01
		Pre-intervention	Follow-up	−2.05	1.48	1.00	−5.96	1.86	
		Post-intervention	Follow-up	−2.27	1.51	1.00	−6.37	1.82	
**PAQ ^a^**	**EG**	Pre-intervention	Post-intervention	−1.095 *	0.04	<0.001	−1.20	−0.99	4.46
		Pre-intervention	Follow-up	−1.044 *	0.03	<0.001	−1.13	−0.96	
		Post-intervention	Follow-up	0.050 *	0.03	0.05	0.00	0.10	
	**CG**	Pre-intervention	Post-intervention	−0.0789 *	0.01	<0.001	−0.12	−0.04	0.92
		Pre-intervention	Follow-up	−0.1215 *	0.02	<0.001	−0.16	−0.08	
		Post- intervention	Follow-up	−0.0427 *	0.01	0.01	−0.08	−0.01	
**Screen time**	**EG**	Pre-intervention	Post-intervention	57.92 *	5.97	<0.001	40.51	75.33	1.38
		Pre-intervention	Follow-up	77.80 *	10.19	<0.001	48.62	106.98	
		Post-intervention	Follow-up	19.88	9.84	0.22	−5.47	45.24	
**Total** **SB time** ** ^a^ **	**CG**	Pre-intervention	Post-intervention	20.76 *	6.55	0.01	3.14	38.39	0.20
		Pre-intervention	Follow-up	8.24	6.80	0.68	−8.05	24.53	
		Post-intervention	Follow-up	−12.52	7.40	0.36	−30.99	5.95	
**Screen time ^a^**	**EG**	Pre-intervention	Post-intervention	95.23 *	9.39	<0.001	68.32	122.14	0.98
		Pre-intervention	Follow-up	119.33 *	20.43	<0.001	62.69	175.98	
		Post- intervention	Follow-up	24.11	18.89	0.61	−21.12	69.33	
	**CG**	Pre-intervention	Post-intervention	34.20 *	10.43	0.01	6.68	61.72	0.16
		Pre-intervention	Follow-up	10.54	10.78	0.66	−13.62	34.71	
		Post-intervention	Follow-up	−23.65	12.31	0.27	−55.37	8.06	
**Total SB time**	**EG**	Pre-intervention	Post-intervention	121.28 *	17.88	<0.001	69.86	172.70	0.75
		Pre-intervention	Follow-up	180.62 *	42.26	<0.001	66.94	294.31	
		Post-intervention	Follow-up	59.34	41.78	0.32	−34.58	153.26	
	**CG**	Pre-intervention	Post-intervention	6.09	22.81	0.79	−38.63	50.81	0.52
		Pre-intervention	Follow-up	−54.93 *	16.67	0.01	−98.92	−10.95	
		Post-intervention	Follow-up	−61.03	25.23	0.08	−126.02	3.97	

SE = Standard error, * significant score at (*p* < 0.05), CI = Confidence interval is based on population values, LB = Lower bound, UB = Upper bound, *d* = Cohen effect size, Adjustment for multiple comparisons (Bonferroni), ^a^ = adjusted mean difference using pre-test score as a covariate. EG = The experimental group (68), CG = The control group (70).

**Table 4 ijerph-19-11314-t004:** Pairwise comparisons between both groups at three times for BMI-for-age z-score, waist circumference, physical activity, and screen time and the total sedentary behavior on weekdays and weekends.

Variable	Test	Experimental Group	Control Group	Mean Difference (I–J)	SE	*p*-Value	95% CI for Difference		*d*
							LB	UB	
**BAZ ^a^**	Pre-intervention	EG	CG	0.01	0.02	1.00	−0.04	0.06	0.35
	Post-intervention ^a^	EG	CG	0.07	0.03	0.28	−0.02	0.16	0.34
	Follow-up ^a^	EG	CG	0.00	0.00	1.00	−0.01	0.01	0.36
**WC**	Pre-intervention	EG	CG	−2.11	1.51	1.00	−6.10	1.88	0.22
	Post-intervention	EG	CG	−2.72	1.56	1.00	−7.28	1.84	0.25
	Follow-up	EG	CG	2.05	1.48	1.00	−1.86	5.96	0.26
**PAQ ^a^**	Pre-intervention	EG	CG	−0.0317 *	0.01	0.01	−0.06	−0.01	0.87
	Post-intervention ^a^	EG	CG	−1.048 *	0.04	0.00	−1.18	−0.92	4.31
	Follow-up ^a^	EG	CG	−0.954 *	0.04	0.00	−1.06	−0.85	4.97
**Screen time (weekdays)**	Pre-intervention	EG	CG	0.99	7.51	0.90	−13.73	15.71	0.02
	Post-intervention	EG	CG	38.15 *	8.95	<0.001	13.67	62.62	0.68
	Follow-up	EG	CG	70.55 *	8.52	<0.001	45.92	95.18	1.37
**Total SB time ^a^ (weekdays)**	Pre-intervention	EG	CG	−5.91	5.43	0.83	−18.91	7.09	−0.18
	Post-intervention ^a^	EG	CG	60.85 *	17.08	<0.001	14.15	107.57	0.45
	Follow-up ^a^	EG	CG	135.95 *	21.19	<0.001	74.71	197.19	1.09
**Screen time ^a^ (weekends)**	Pre-intervention	EG	CG	26.19 *	7.35	<0.001	6.42	45.96	0.61
	Post-intervention ^a^	EG	CG	87.22 *	13.71	<0.001	48.73	125.72	1.15
	Follow-up ^a^	EG	CG	134.98 *	18.63	<0.001	82.11	187.86	1.69
**Total SB time** **(weekends)**	Pre-intervention	EG	CG	58.46	24.21	0.08	−3.82	120.73	0.39
	Post-intervention	EG	CG	173.65 *	34.93	<0.001	76.80	270.50	0.80
	Follow-up	EG	CG	294.015 *	35.55	<0.001	190.44	397.58	1.38

SE = Standard error of the sample, CI = Confidence interval is based on population values, LB = Lower bound, UB = Upper bound, *d*= Cohen effect size, *n* = Number of respondents, EG = The experimental group (68), CG = The control group (70), * *p*-value significant at <0.05; Adjustment for multiple comparisons (Bonferroni), ^a^ = adjusted mean difference using pre-test score as a covariate.

**Table 5 ijerph-19-11314-t005:** Comparison of body image satisfaction between groups and within time for both groups.

Time		EG *n* (%)	CG *n* (%)	Between Groups	
				χ^2 a^	*p*-Value
**Pre-intervention**		11 (14.5)	24 (30.8)	5.820 *	0.016
**Post-intervention**		44 (57.9)	26 (33.3)	9.366 *	0.002
**Follow-up**		39 (51.3)	23 (29.5)	7.626 *	0.006
**Within group**	q value ^b^	49.947 *	2.800		
	*p*-value	<0.001	0.247		

*n* = Number of respondents, EG = The experimental group (68), CG = The control group (70), ^a^ = Chi-square test, ^b^: Cochran’s Q test, * Significant score at (*p* < 0.05).

## Data Availability

The data from this study can be acquired from the corresponding author upon request.

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
