# Peer review of "The Effects of a Physical Activity, Nutrition, and Body Image Intervention on Girls in Intermediate Schools in Saudi Arabia"

_ijerph, 2022, doi:10.3390/ijerph191811314_

Round 1
Reviewer 1 Report
2.3. Educational Intervention
The presentation of the research methodology related to the six educational interventions, designed by the authors, is much too generally addressed in the article. Thus, (for example), the allocated time/session (out of the total time of 90 min.) is not specified for each individual component, respectively: physical activity, nutrition and body image. In addition, the methods of grading and combining the topics on these three directions/sessions are not presented. At the same time, the authors mention that each session consisted of a theoretical section and one of practical activities, without specifying what these activities consist of for each direction (physical activity, nutrition, body image), as well as the time allocated for each of them /educational (training) session.
2.5.2. Physical Activity
The original physical activity questionnaire for older children (PAQ-C) contains 10 items. The authors present only nine and say that they have adapted the items. Which item was eliminated/removed and what was the adaptation of the other nine items?
3. Results
I would suggest that the tabular data collected and processed for the three evaluation moments (pre-intervention, post-intervention and follow-up), be represented graphically, to highlight more clearly the impact of the intervention on the experimental group, compared to the control group.
References
I recommend that the references be reviewed because they do not fully meet the requirements. No DOI was included for references where it exists.
Author Response
Dear Reviewer
Thank you for giving me the opportunity to submit a revised draft of my manuscript entitled “The Effects of a Physical Activity, Nutrition and Body Image Intervention on Girls in Intermediate Schools in Saudi Arabia”. We are grateful to the reviewers for their insightful comments on our paper.
Attached is a file contains a point-by-point response to the reviewers’ comments and concerns.
We look forward to hearing from you in due time regarding our submission.
Sincerely,
Kind regards

Reviewer 2 Report
This is a fine report about an interesting field.
The paper is well written and references seem updated.
Nevertheless, I have some concerns:
1. There should be no abbreviations in the abstract. Please remove.
2. In the introduction, lines 34 through 48 are more appropriate for discussion.
· The introduction should be more extensive.
3. Why was one school the control group and the other the experimental group? The level of education could have been different between schools, it would have been more appropriate to divide the grupo between schools.
4. Verse 113 from "Both" to 115 please delete.
5. Statistical analysis should be described in more detail.
6. Please standardize the values in parentheses (p<0.05) rather than (p < 0.05).
Author Response

(The authors gave the same response as above.)
